# A Choreography-Based and Collaborative Road Mobility System for L'Aquila City

**Marco Autili** [ID]**, Amleto Di Salle** [ID]**, Francesco Gallo** [ID]**, Claudio Pompilio** [ID] **and Massimo Tivoli \*** [ID]

Department of Information Engineering, Computer Science and Mathematics, Università degli Studi dell'Aquila, 67100 L'Aquila, Italy; marco.autili@univaq.it (M.A.); amleto.disalle@univaq.it (A.D.S.); francesco.gallo@univaq.it (F.G.); claudio.pompilio@univaq.it (C.P.)

**\*** Correspondence: massimo.tivoli@univaq.it

**Abstract:** Next Generation Internet (NGI) is the European initiative launched to identify the future internet technologies, designed to serve the needs of the digitalized society while ensuring privacy, trust, decentralization, openness, inclusion, and business cooperation. NGI provides efficient support to promote diversity, decentralization and the growth of disruptive innovation envisioned by smart cities. After the earthquake of 6 April 2009, the city of L'Aquila is facing a massive and innovative reconstruction process. As a consequence, nowadays, the L'Aquila city can be considered as a living laboratory model for applications within the context of smart cities. This paper describes and evaluates the realization of a Collaborative Road Mobility System (CRMS) for L'Aquila city by using our CHOReVOLUTION approach for the automated choreography production. The CRMS allows vehicles and transport infrastructure to interconnect, share information and use it to coordinate their actions.

**Keywords:** smart cities; distributed systems; choreography; services oriented systems

## 1. Introduction

The commitments listed in the *Urban Agenda 2030* [1], endorsed at the European level in the *Pact of Amsterdam* (Urban Agenda for the EU) [2] for sustainable urban development are based on three dimensions: economic, social and environmental, in a balanced and integrated manner. The challenge is to promote the sharing of possibilities and benefits that urbanization can offer ensuring a full and productive employment for all, through the sustainable management of natural resources in cities and human settlements.

Smart City has been identified as being an exemplary example of a response to address the current and future complex challenges listed in Urban Agenda [3]. Albino et al. describe the smart city as an abstract projection of the future communities, an application and a conceptual boundary defined by a set of needs that find answers in technologies, services and applications related to different domains: smart building, inclusion, energy, environment, government, living, mobility, education, health, and much more [4–6]. The aim is the construction of a new kind of community, a great technological and immaterial infrastructure that brings people and objects together, integrating information, generating intelligence, producing inclusion and improving our daily lives [7].

One of the key technological infrastructures enabling Smart Cities is the Internet network [8]. The Internet has emerged to a critical infrastructure for society and economy as a whole, similar to any other utility, e.g., infrastructure for electricity and water supply. Indeed, through digitalization and the help of the Internet, all our major activities such as living, working, engaging in hobbies, moving, and consuming take place in various combinations of physical and digital forms. The Internet will

be(is) the mother-ship for "prosumption" of digital contents, based on information and knowledge generation, sharing, and elaboration.

Although the internet has led to the development of a new digital culture, it has been designed for fixed terminals, and therefore it shows an inefficient behaviour for mobile, heterogeneous and nomadic terminal (or "smart people") [9]. It is necessary to redefine both conceptually and concretely the Internet infrastructure, in order to provide a generic and efficient support to terminals and mobile applications. This should foster diversity, decentralization and the growth of disruptive innovation. This new vision has been formally identified as Next Generation Internet (https://www.ngi.eu) (NGI). NGI is the European initiative launched to identify the future internet technologies, designed to serve the needs of the digitalized society while ensuring privacy, trust, decentralization, openness, inclusion, and business cooperation. The NGI will be built on addressing technical challenges across all levels (top-down and bottom-up) of Internet usage (user, application, management, infrastructure) to support a free, open and interoperable Internet developed and used at its full potential at all levels of technology, business and society [10].

This boosts the trend towards distributed and edge computing [11], where resources are not located en-masse in one location, but spread over a wide area. The degree of infrastructure decentralisation ranges from fully centralised to distributed, reflecting the increasing influence of service-oriented systems and IoT devices. Indeed, systems are increasingly produced by integrating existing software and devices, and *reuse-based* software engineering is becoming the main development approach for building business and commercial systems [12,13]. Services play a central role in this vision as an effective means to achieve interoperability among parties of a business process, and new systems can be built by reusing and composing existing services and things.

Service choreographies support the reuse-based service-oriented philosophy in that they represent a powerful and flexible approach to realize systems by (possibly) reusing services and devices, and composing them in a fully distributed way. The need for choreographies was recognized in the Business Process Modeling Notation 2.0 (http://www.omg.org/spec/BPMN/2.0.2/) (BPMN2), which introduced *Choreography Diagrams* to offer choreography modeling constructs. During the last decade, with the aim of bringing the adoption of choreographies to the development practices adopted by IT companies, we focused our research and development activity on practical and automatic approaches to support the realization of reuse-based service choreographies [12,14–22]. This activity has been mainly funded by two EU projects: the FP7 CHOReOS and its follow-up H2020 CHOReVOLUTION (www.choreos.eu—www.chorevolution.eu).

The city of L'Aquila, after the devastating earthquake of 6 April 2009, is facing a massive reconstruction process, which has acquired an even more challenging connotation in the face of new technological and social challenges that are becoming real urban regeneration. In this context, the INCIPICT (http://incipict.univaq.it) (INnovating City Planning through Information and Communication Technologies) project aims at creating a metropolitan network consisting of a fiber optic ring dedicated to the Public Administration, Schools, and Universities. This infrastructure enables the NGI vision, and hence it can be exploited to provide innovative services for citizens.

This paper presents a case study that has been developed within one of the INCIPICT application scenarios and that employs the CHOReVOLUTION approach to realize a Collaborative Road Mobility System (CRMS) for L'Aquila city. The CRMS allows vehicles and transport infrastructure to interconnect, share information and use it to coordinate their actions. Furthermore, it provides traffic coordination services exploited through a mobile app for assisting drivers in the most eco-friendly and comfortable driving experience [9,23].

The paper is structured as follows. Section 2 discusses related work. Section 3 overviews the CHOReVOLUTION approach by discussing the choreography realizability enforcement problem it solves, and by describing underlying technologies in Section 3.1. Section 4 presents the CRMS case study, and Section 4.1 describes a realization of it obtained by using the CHOReVOLUTION approach

at work. Section 4.2 reports the outcomes of the experimental evaluation performed on the CRMS case study. Conclusions and future research directions are given in Section 5.

## 2. Related Work

The approach described in this paper is mainly related to the automated choreography realization.

VerChor is a framework for choreography design and verification [13]. The framework translates the existing choreography languages, i.e., conversation protocols or BPMN 2.0 choreographies, into the LOTOS NT (LNT) process algebra [24] by means of a choreography intermediate format (CIF). In this way, the framework exploits the analysis techniques provided by the CADP [25] toolbox to check a set of key properties that choreographies must respect for ensuring correctness of the system under development. Our approach leverages on theoretical results exploited in [13] to provide practical and automatic support for the realization of reuse-based service choreographies and hence to boost the adoption of choreographies in the development practices adopted by IT companies.

Güdemann et al. [26] propose a method to enforce synchronizability and realizability of a choreography. The method generates monitors that, similarly to our notion of CD, act as distributed controllers. However, the synthesis technique is different from ours in that monitors are generated by iteratively refining their behavior.

Basu et al. [27] identify a class of systems where choreography conformance can be efficiently checked even in the presence of asynchronous communication. This is done by checking synchronizability. The approach characterizes relevant properties to check choreography realizability that represents the starting point for choreography realizability enforcement. However, it is focused on a fundamentally different problem from ours. It statically checks realizability and does not account for its automatic enforcement at run time, including code synthesis, actual deployment and execution.

The ASTRO toolset supports automated composition of services [28]. It aims to compose a service out of a business requirement and the description of available external services. More specifically, a planner component automatically synthesizes the code of a centralized process that achieves the business requirement by interacting with the available external services. Unlike our approach, ASTRO deals with orchestration-based processes rather than decentralized choreography-based ones.

The CIGAR (Concurrent and Interleaving Goal and Activity Recognition) framework is for multigoal recognition [29]. It decomposes an observed sequence of multigoal activities into a set of action sequences specifying whether a goal is active in a specific action. Goal recognition concerns learning a model of an agent by observing the agent's actions while interacting with the environment. In contrast, realizability enforcement decentralizes the coordination logic specified by choreography.

Carbone et al. [30] present a unified programming framework for developing choreographies that ensure deadlock freedom and communication safety. Developers design both protocols and implementation from a global perspective. Correct endpoint implementations are then automatically generated. Lanese et al. [31] discuss some of the extensions of the Jolie orchestration language. Differently from our approach, they focus on developing choreographies from scratch, rather than realizing them through service reuse.

Passerone et al. use a game theoretic approach for checking whether incompatible component interfaces can be made compatible by inserting a converter between them which satisfies specified requirements. This approach is able to automatically synthesize the converter [32]. In contrast to our method, their method needs as input a deadlock-free specification of the requirements that should be satisfied by the adapter, by delegating to the user the non-trivial task of specifying that. Our validation phase, automatically achieve deadlock freeness in the choreography specification.

Bennaceur and Issarny presented an approach that, exploiting ontology reasoning and constraint programming, allows for automatically inferring mappings between components interfaces [33]. Importantly, these mappings guarantee semantic compatibility between the operations and data. Although valuable and powerful, this approach does not account for development effort whose

aim is to bring the adoption of choreographies to the development practices currently adopted by IT companies.

## 3. CHOReVOLUTION Approach

This section introduces the CHOReVOLUTION approach to address the automatic realizability enforcement problem in the context of choreography-based service-oriented systems by reusing third-party services.

When considering choreography-based service-oriented systems, the following two problems are usually considered: (i) *realizability check*—checks whether the choreography can be realized by implementing each participant so that it conforms to the played role; and (ii) *conformance check*—checks whether the set of services satisfies the choreography specification. In the literature, many approaches have been proposed to address these problems, e.g., [13,27,34–40].

However, to put choreographies into practice, we must consider realizing choreography-based service-oriented systems by reusing third-party services. This leads to a further problem: the *automatic realizability enforcement*. It can be informally phrased as follows: *given a choreography specification and a set of existing services, externally* **coordinate** *and* **adapt** *their interaction so to fulfill the collaboration prescribed by the choreography specification*.

In our previous works [18,20,22,41,42], we describe how additional software entities—called **Coordination Delegates** (CDs)—can be automatically synthesized to proxify and coordinate the participants' interaction. When needed, CDs are interposed among the participant services according to a suitably generated architecture description. CDs enforce the collaboration prescribed by the choreography and they do not require any effort to developers concerning coordination issues. Our approach spares developers from writing code that goes beyond the realization of the business logic internal to those (single) choreography tasks for which no existing applications or services can be reused. Developers just need to fill-in-the-blank of automatically generated code templates.

For what concerns adapting the interaction among the participant services (if needed), in our previous works [16,21,43], we describe how other additional software entities—called **Adapters (As)**—are automatically synthesized to solve possible interaction protocol mismatches between the interaction protocol of the abstract participant services, as described by the choreography specification, and the interaction protocol of the existing concrete services that are bound to the abstract ones. The Adapters are generated by exploiting a representation of the "gap" between the protocol of the abstract participants and the one of the concrete participants given in terms of (possibly complex) data mappings over both operation names and I/O messages, and their flows.

Figure 1 shows a simple example of a BPMN2 choreography diagram, the standard de facto for specifying choreographies. Choreography diagrams define the way business participants coordinate their interactions. The focus is on the exchange of messages among the involved participants. A choreography diagram models a distributed process specifying activity flows where each activity represents a message exchange between two participants. Graphically, a choreography task is denoted by a rounded-corner box. The two bands, one at the top and one at the bottom, represent the participants involved in the interaction captured by the task. A white band is used for the participant initiating the task that sends the initiating message to the receiving participant in the dark band that can optionally send back the return message.

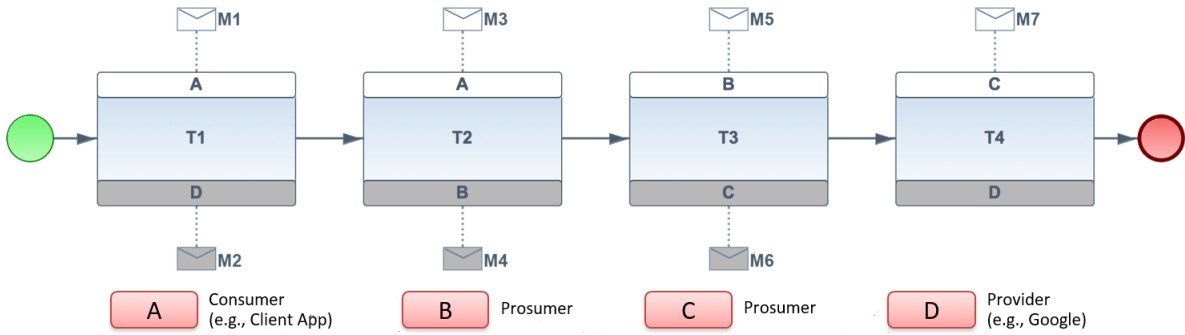

**Figure 1.** BPMN2 choreography diagram example.

The choreography in Figure 1 involves four participants, A, B, C, and D, for the execution of four sequential tasks, T1, T2, T3 and T4. Specifically, A sends the message M1 to D, enabling it for the execution of T1. After that, D replies to A by sending the message M2. At this point, A sends M3 to B that, after the execution of T2, replies M4 to A and sends M5 to C. Only when M5 is received by C, it executes T3, replies M6 to B and sends M7 to D. Finally, D executes T4 and the choreography ends.

By analyzing the choreography, we can distinguish three different types of participants: *consumer*, *provider*, and *prosumer* (i.e., both consumer and provider). For instance, the consumer participant A might be played by an existing Client App; the provider participant D by an existing Web Service, e.g., Google Maps; B and C might be two prosumers that have to be developed from scratch in order to realize the specified choreography.

Figure 2 shows the "most general" architectural configuration of the system that realizes the choreography specified in Figure 1. The top-most layer contains the services representing the business logic. In particular, a:A denotes that the role of the consumer participant A is played by a, the Client App in our example; d:D denotes that the role of the provider participant D is played by d, an existing provider service to be reused, whereas, concerning the participant B and C, in the figure, we do not make use of the notation x:X simply to indicate that they are not existing prosumer services and thus they can be either implemented from scratch or partially reused (for the provider part). Then, the second layer contains the **Binding Components** (BCs) [44] that cope with middleware-level protocol adaptation, e.g., REST versus SOAP adaptation. It is worth mentioning that SOAP is the default interaction paradigm for the underlying layers. Finally, the last two layers include the Adapter and CD artefacts for adaptation and coordination purposes, respectively.

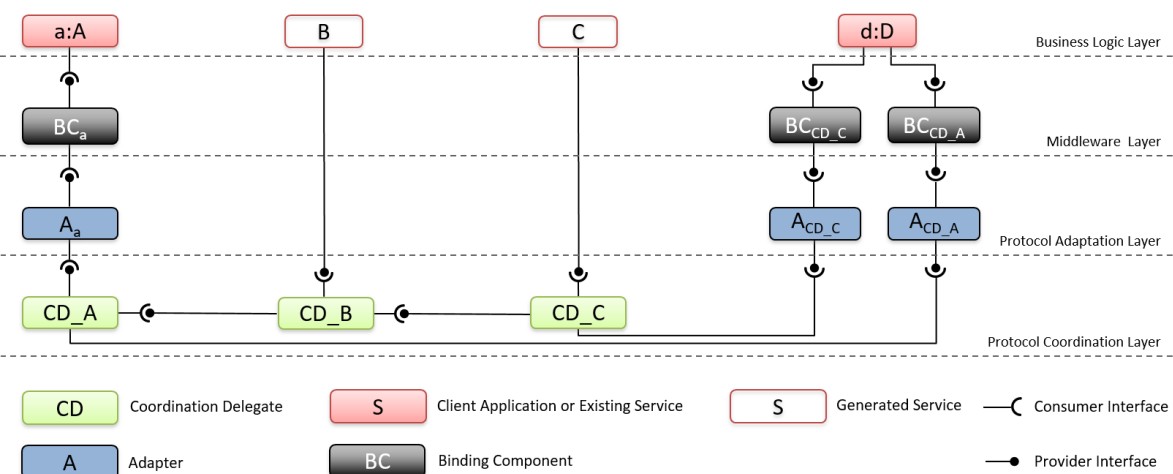

**Figure 2.** Choreography architectural style (a sample instance of).

The additional artefacts are not always required; rather, it depends on the specified choreography and the characteristics (e.g., application-level interaction protocols, interface specifications,

middleware-level interaction paradigms) of the existing services that have been selected to instantiate the roles of the choreography participants.

To address the *automatic realizability enforcement* problem, we have defined the CHOReVOLUTION synthesis process that allows for realizing dynamic choreographies via distributed coordination of services. Figure 3 shows an overview of the synthesis process. It takes as input the choreography specification—given in terms of BPMN2 Choreography Diagram and Messages XML Schema—and a set of third-party services used as possible candidates to play the roles of the choreography abstract participants. From these inputs, the synthesis process automatically generates all the needed additional software entities, i.e., CDs for solving coordination issues, Adapters for adapting application-level interaction protocols, and BCs for adapting middleware-level interaction protocols. At the end of the process, it is also generated the Choreography Deployment Description (called `ChorSpec`). The ChorSpec is an XML file referring to all the generated artefacts in order to specify their dependencies. In addition, it contains information required to correctly perform the choreography deployment on the Cloud (bottom layer in Figure 3).

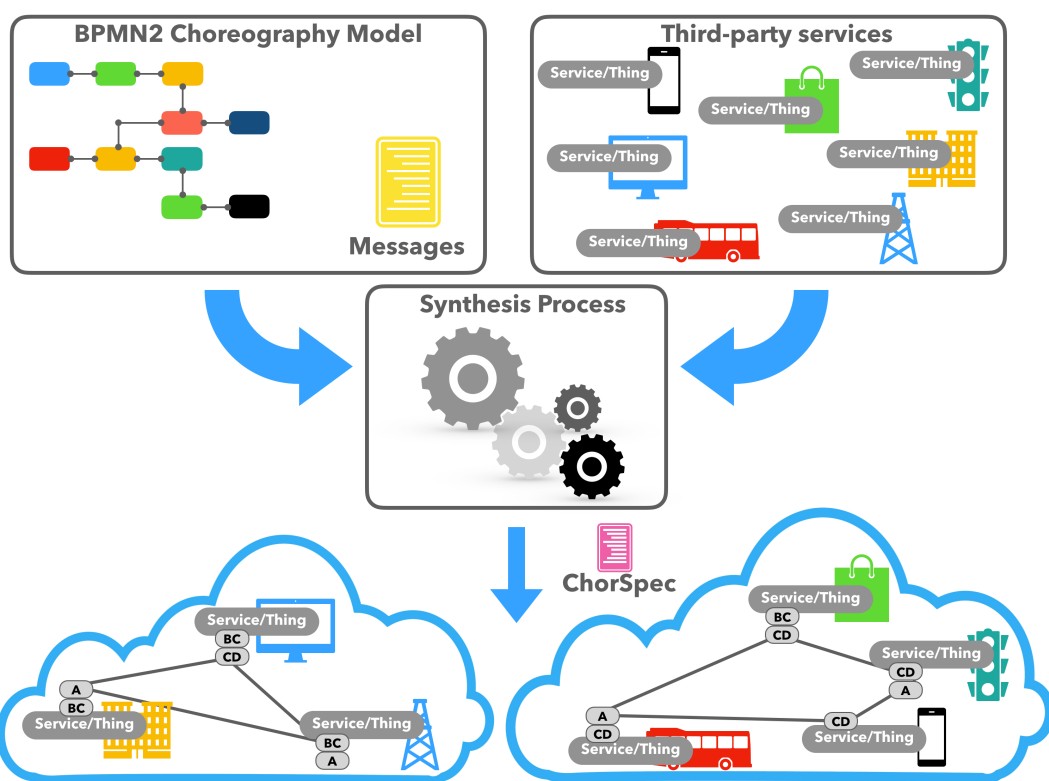

**Figure 3.** CHOReVOLUTION synthesis process.

The synthesis process described above is supported by an Integrated Development and Runtime Environment (IDRE), named CHOReVOLUTION IDRE. The CHOReVOLUTION IDRE makes the realization of choreography-based applications easier by sparing developers from writing code that goes beyond the realization of the internal business logic.

The IDRE is an open-source and free software, available under the Apache license. It is available as a ready-to-use bundle or you can download each component from http://www.chorevolution.eu/b in/view/Documentation/Download. All the documentation can be found at http://www.chorevolut ion.eu/bin/view/Documentation/WebHome.

### 3.1. CHOReVOLUTION IDRE Underlying Technologies

CHOReVOLUTION IDRE includes software tools for choreography modeling, synthesis, security, identity management and cloud (with monitoring and overall management at run-time).

As depicted in Figure 4, the CHOReVOLUTION IDRE is layered into: (1) a front–end layer; (2) a back–end layer; and (3) a cloud layer. The red boxes in the figure contain the IDRE components developed within CHOReVOLUTION from scratch. In particular, they are: the CHOReVOLUTION Studio, the CHOReVOLUTION Console, and the Synthesis Processor together with the artifacts it generates. As detailed below, the components outside the boxes are the ones developed within CHOReVOLUTION and built on top of existing open-source projects. For instance, the Identity Manager extends the Apache Syncope project [45]. It is worth noticing that the choice about the existing projects the IDRE relies on comes from the partners of the CHOReVOLUTION consortium. However, the IDRE is an extensible platform and, as such, in the future, it may also support other technologies such as: Kubernetes for deployment and enactment, IBM's AIM for identity management.

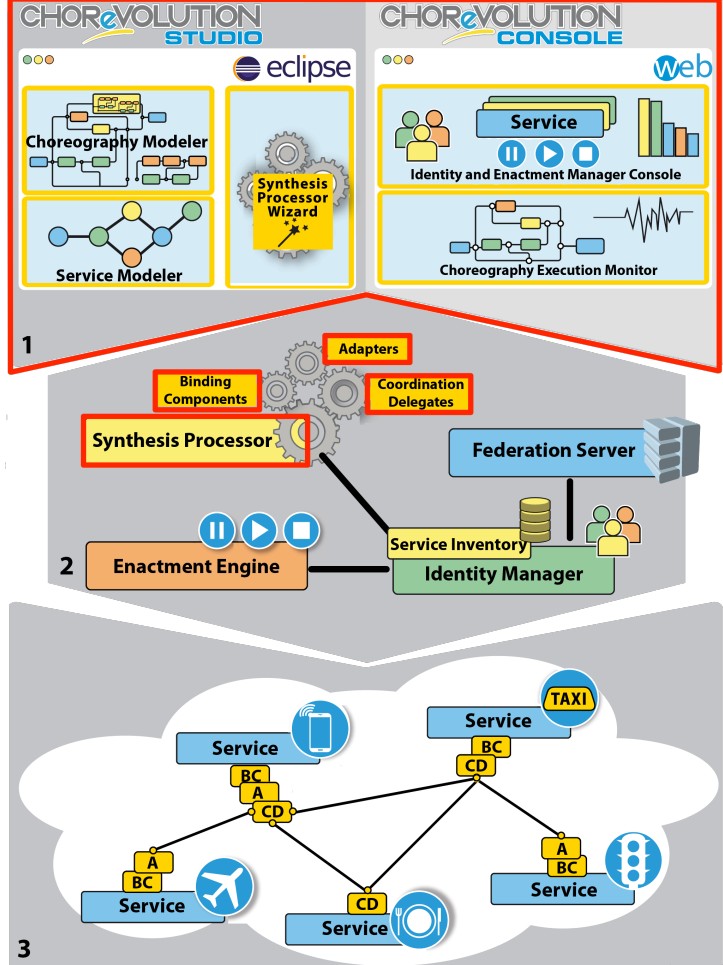

**Figure 4.** CHOReVOLUTION IDRE overview.

**(1) The Front–end layer** consists of the following:

**(1.1)—** The **CHOReVOLUTION Studio** is an Eclipse-based IDE that allows for (i) designing a choreography exploiting BPMN2 Choreography Diagrams; (ii) defining all the details required to instrument the interaction among the services involved in the choreography (e.g., service signatures, identity attributes and roles); and (iii) driving the generation of BCs, As, and CDs exploiting the automated generation facilities offered by the back-end layer.

**(1.2)—** The **CHOReVOLUTION Console** is a web-based application that allows for (i) configuring, administering and triggering actions on running services and choreographies; and (ii) monitoring the execution of a choreography with respect to relevant parameters, such as execution time

of choreography tasks, number of messages exchanged for the execution of tasks, end-to-end deadlines, etc.

**(2) The Back–end layer** consists of the following:

**(2.1)—** The **Synthesis Processor** implements the activities of the synthesis process shown in Figure 3. In particular, it takes as input the BPMN2 choreography diagram and the models of the participant services, and generates all the needed additional software entities that are required to concretely realize the choreography, i.e., CDs, As, and BCs. Finally, it generates a concrete description of the choreography (`ChorSpec`) that is passed to the Enactment Engine (via the Identity Manager) for deployment and enactment purposes.

**(2.2)—** The **Enactment Engine (EE)** is a REST API that extends the Apache Brooklyn project [46]. It automatically deploys the choreography based on the choreography deployment description by using the Cloud Layer. The EE also interacts with the Identity Manager to include into the deployment description the actual deployment and runtime details. Then, once a choreography is deployed and running, the EE listens for command requests from the Identity Manager for runtime choreography control. It is worth noticing that, although choreography monitoring and control is performed by centralized IDRE components (e.g., EE and IdM), the realization and running of the choreography still remain fully distributed into the various artifacts generated by the Synthesis Processor.

**(2.3)—** The **Federation Server** handles the runtime authentication and authorization for services that use different security mechanisms at the protocol level by storing various credentials on behalf of the caller.

**(2.4)—** The **Identity Manager (IdM)** is based on Apache Syncope project [45] and it is responsible for managing users and services. In particular, the IdM is able to query the services for supported application contexts and played roles; force a specific application context for a certain service (put in "maintenance" or disable/enable). The Service Inventory is a sub-component of the IdM. It acts as a central repository for the description models of the services and things that can be used during the synthesis process.

**(3) The Cloud layer** executes concrete service choreography instances on a cloud infrastructure and adapts their execution based on the actual application context. At execution time, for each choreography, in the CHOReVOLUTION cloud, there are (i) a set of choreography instances at different execution states; and (ii) a set of virtual machines executing a custom-tailored mix of services and middleware components to serve different parts of the choreography. VMs are installed and configured with services according to selectable policies. Due to the fact that EE is based on Apache Brooklyn, the CHOReVOLUTION IDRE is not constrained to a specific Infrastructure as a Service (IaaS) platform (e.g., Open Stack [47], Amazon EC2 [48]).

The IDRE mainly targets three types of users described as follows:

**Service providers** interact with the CHOReVOLUTION Studio to define the description models (i.e., interface and security models) of existing services and then publish them into the Service Inventory. The benefit they obtain is to foster and ease the reuse of their services by developers, hence increasing the opportunities to be involved in new businesses.

**Choreography developers** interact with the CHOReVOLUTION Studio to (i) model a choreography by using the Choreography Modeler; (ii) Realize the modeled choreography through the automatic synthesis of BCs (for solving heterogeneity issues), CDs (for solving coordination issues) and As (for solving interface mismatches). This is done by exploiting the Synthesis Processor.

**Choreography operators** interact with the CHOReVOLUTION console to (i) deploy and enact the generated choreography-based application through a structured process that involves the back-end

layer; (ii) monitor the status of the execution cloud environment; and (iii) monitor the execution of the choreography instances and managing their lifecycle.

## 4. Case Study

At the moment of writing, ten years after the earthquake, L'Aquila represents a challenging context. It was 3:32 a.m. on 6 April 2009 when an earthquake struck the city of L'Aquila, causing death and destruction [49]. After this devastating event, L'Aquila has started a long but inexorable process of social, physical and economic reconstruction that has allowed it to receive the SMAU (Smart Communities Milano 2015) award for the project "L'Aquila Smart City: Rete Elettrica Intelligente e Mobilità Verde" (Smart Electric Grid and Green Mobility) [50]. The operations in progress will make the new electricity grid more secure and efficient and allow for connect to the network the private ecological equipment for the production of energy, allowing for redistributing this energy in case of blackouts. Taking advantage of the broadband designed for the electricity network and the reconstruction work already in place, the project aims to diffuse the optical fiber extensively, touching all buildings, and hence increasing the speed of information exchange via the Internet. Finally, the installation of charging stations for electric vehicles and awareness-raising activities aimed at citizens, car manufacturers and local administrations wants to facilitate the diffusion of sustainable mobility in the city to allow citizens to move over the wide area without weighing on the environment, reducing emissions of harmful gases and using fossil fuels.

With the opportunity to interpret reconstruction in an innovative way and the absence of conditions hindering the widespread deployment of technological solutions at the base of a Smart City [51], the INCIPICT project, coordinated by the University of L'Aquila and launched in late 2014 [52], supports the economic development of the city and improves the quality of life of citizens through a widespread use of Information and Communication Technologies (ICTs). The project aims to create a Living Laboratory model, leveraging on the reconstruction of buildings and the radical re-organization of the city's sub-services network (water supply, electricity supply, telecommunications networks), to create essential and enabling components for innovative urban development and support for research activities. These innovative services will be supported concretely through a project of experimentation of the 5G technology [9]. The project is promoted by the Ministry of Economic Development (MISE) [53], which will be carried out in partnership with Open Fiber (https://openfiber.it/en), a joint venture of Enel (https://www.enel.com) and Cassa Depositi e Prestiti (https://en.cdp.it), Wind Tre [54] and ZTE (http://www.zteitalia.it).

Our use case belongs to this highly challenging context. In particular, it realizes a Collaborative Road Mobility System (CRMS) for L'Aquila city. The aim is to realize a system that allows vehicles and transport infrastructure to interconnect, share information and use it to coordinate their actions. The CRMS provides traffic coordination services exploited through a mobile app for assisting drivers in the most eco-friendly and comfortable driving experience [9]. Referring to the Smart Mobility indicator defined in [55], related system key objectives, and foundational mobility model concepts in [56], our CRMS represents a mobility system in its "intermediate phase" of implementation. In fact, it is a pilot project collocated within the context of the ongoing INCIPICT project leveraging a work-in-progress optical network infrastructure. Its key objectives are the reduction of air pollution and traffic congestion. In the future, with the adoption of 5G technology in the city of L'Aquila, CRMS will move to a mature phase.

Before describing the CRMS case study in detail, it is worth mentioning that having a mobility system in its intermediate phase is the best that we can achieve in the partial (and under evolution) execution and infrastructural environment offered today by the L'Aquila city. Furthermore, for the CRMS use case, differently from other INCIPICT use cases, emergency management and disaster recovery issues are left for future work.

The CRMS use case includes two choreographies: Eco Driving Planning Application (EDP) and Smart Data Collection Application (SDC).

Figure 5 shows EDP choreography specification by using a BPMN2 Choreography Diagram. The application is triggered when driver starts the navigation app, i.e., `Driver` and inputs the origin and destination information that are then sent to `Eco Route Planner` (choreography task `Get Eco Routes`). Starting from this information, `Eco Route Planner` retrieves a set routes (choreography task `Get Routes`) and it initiates two parallel flows (see the parallel branch represented as a rhombus marked with a "+", with two outgoing arrows, namely a Diverging Parallel Gateway, just after the choreography task `Get Routes`) to collect, for each route, environmental and traffic information. The resulting information together with the route information are sent to the `Driver` (choreography task `Set Eco Routes`).

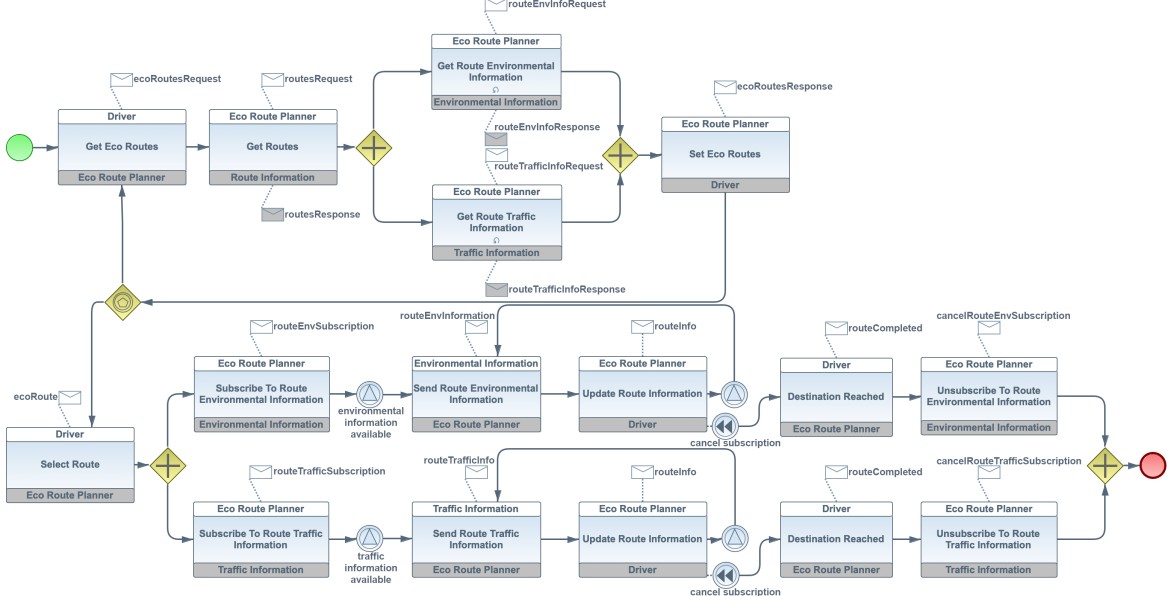

**Figure 5.** Eco Driving Planning choreography.

After selecting a route, the driver is notified when new environmental or traffic information is available. This is done through two parallel flows in which `Eco Route Planner` subscribes to the environmental and traffic information, respectively. An event notifies the availability of new environmental or traffic information for a specific route (see the intermediate event represented as a circle with a triangle inside). Then, the new information is transmitted to the `Eco Route Planner` that, in turn, sends the related updated route information to the `Driver`. When the driver reaches the destination, this is notified to the `Eco Route Planner` (see the intermediate event represented as a circle with two leftwards arrows inside) that unsubscribes to the environmental and traffic information updates.

The Smart Data Collection choreography (Figure 6) is a long-running choreography. It is designed to continuously collect environmental and traffic information and use it for fast response upon requesting. The choreography starts with the `Viability Control Center` sending the area of interest to the `Smart Data Collector`. From this information, `Smart Data Collector` starts two parallel executions to retrieve the related environmental and traffic information. In particular, the top-most parallel branch concerns the environmental information. In this branch, the `Environmental Information` participant receives weather and air quality information after subscribing to it. `Environmental Information` can possibly unsubscribe from the reception of this information. The bottom-most parallel flow has the same structure. It involves the retrieval of traffic information through the subscription to congestion, accident and road status information.

We used the choreographies specification in Figures 5 and 6 to realize a CRMS for the city of L'Aquila, where local municipalities offer a number of publicly available services to be reused. They have been reused—as black-box third-party software—to instantiate the roles of the participants

`Route Information, Weather, Air Quality, Congestion, Accidents,` and `Road Status`. The other participants had to be developed from scratch, and here our choreography synthesis method comes into play. These participants represent the missing logic to be composed and coordinated with the logic offered by the reused services. As formalized in [20], due to the distributed nature of the system, when realizing all the nested parallel and branching flows specified by the choreography, coding the missing logic by hand and coordinating its interaction with the logic coming from the reused services is non-trivial and error prone. Thus, automated support is desirable.

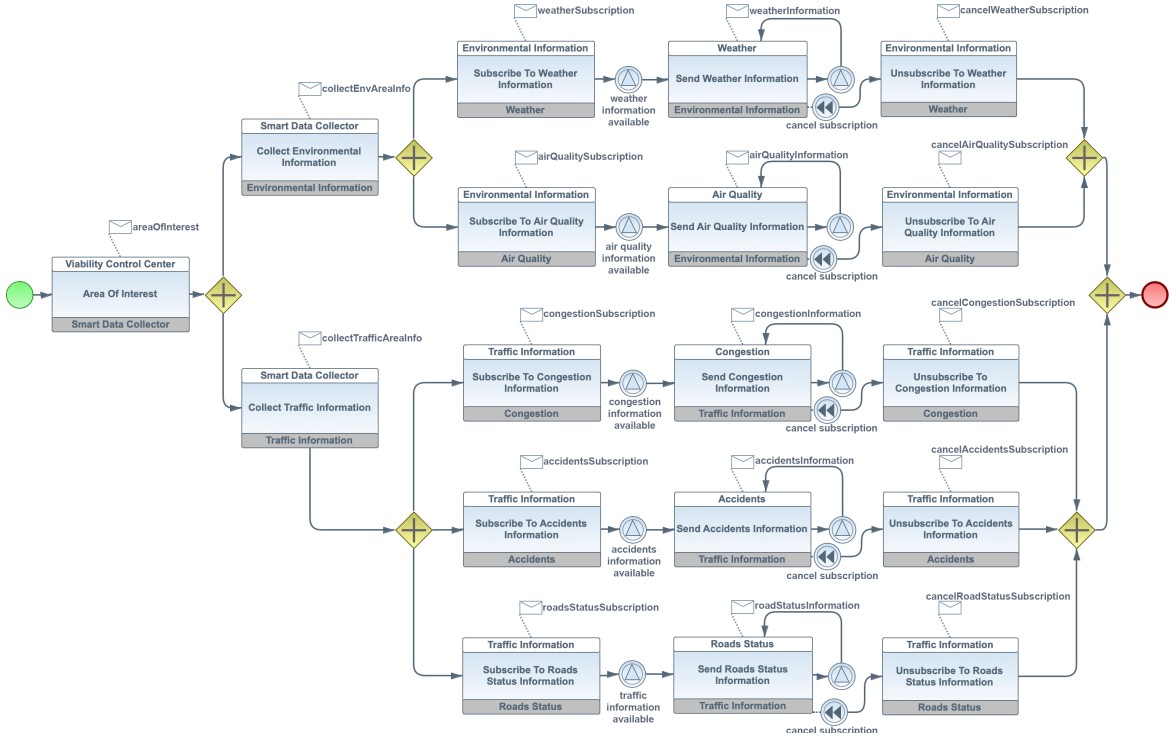

**Figure 6.** Smart Data Collection choreography.

*4.1. The CHOReVOLUTION Synthesis Method at Work on the CRMS Use Case*

This section describes how the CHOReVOLUTION synthesis method can be exploited to realize the CRMS use case in the context of L'Aquila city. The following is a description of the services that play the roles of the participants: `Roads Status, Weather, Air Quality, Congestion` and `Accidents`.

**– Roads Status** provides real-time information about the roads status by combining several kinds of informations including data gained from liquid presence sensors (by 5G technologies) dislocated along the main streets of the city, and on the main communication roads that connect the city with the periphery.

**– Weather** provides weather information by using data obtained from a local source, i.e., the Center of Excellence Telesensing of Environment and Model Prediction of Severe events (CETEMPS) (http://cetemps.aquila.infn.it). The CETEMPS manages a considerable network of instrumentation and data from remote sensing (ground and satellite data): a primary station for receiving second generation Meteosat data (https://www.eumetsat.int/website/home/index.html), a meteorological radar, a wind profiling radar, an acoustic radar, optical radars, a station for radio survey and ozone survey, weather stations, induced fluorescence detectors, radiometers and photometers.

**– Air Quality** provides information on air quality, which is assessed by monitoring concentrations of pollutants by sensors deployed at specific points in the city. These analyses are accompanied by the study of meteorological parameters that affect the dispersion of pollutants (wind speed and

direction, humidity, radiation, etc.). This service can be used to trace low-polluting routes or, in the case of multi-modal transport, which involves the use of alternative transportation vehicles such as electric bicycles, electric scooters, buses, cars, etc., to trace routes that allow for avoid areas with high pollution. It uses data gained from the Regional Environmental Protection Agency (ARTA) (https://www.artaabruzzo.it) that manages the regional network ("Regione Abruzzo") for monitoring air quality.

**– Congestion and Accidents** exploits a traffic congestion monitoring system using real traffic data. The system consists of a vehicle detector (VD) sensors subsystem and a control center subsystem. The VD sensor subsystem is used to collect traffic data and send it to the control center.

These services are REST [57] services and hence they are constrained to a client-server architectural style that is not suitable for the event-driven communication required by the CRMS use case. Thus, for each service, we developed a new service that implements the publish–subscribe pattern through the Message Queue Telemetry Transport (MQTT) (http://mqtt.org). In particular, each newly developed service interacts with the related service and, whenever new information is available, it is sent to the corresponding subscribers. Moreover, as a route provider (`Route Information` participant) for the Eco Driving Planning choreography, we have chosen Google Maps (https://cloud.google.com/maps-platform/). Starting from these services, and from specification of the Eco Driving Planning and Smart Data Collection choreographies (see previous section), our synthesis method produces the related choreography-based systems, whose architectures are shown in Figures 7 and 8.

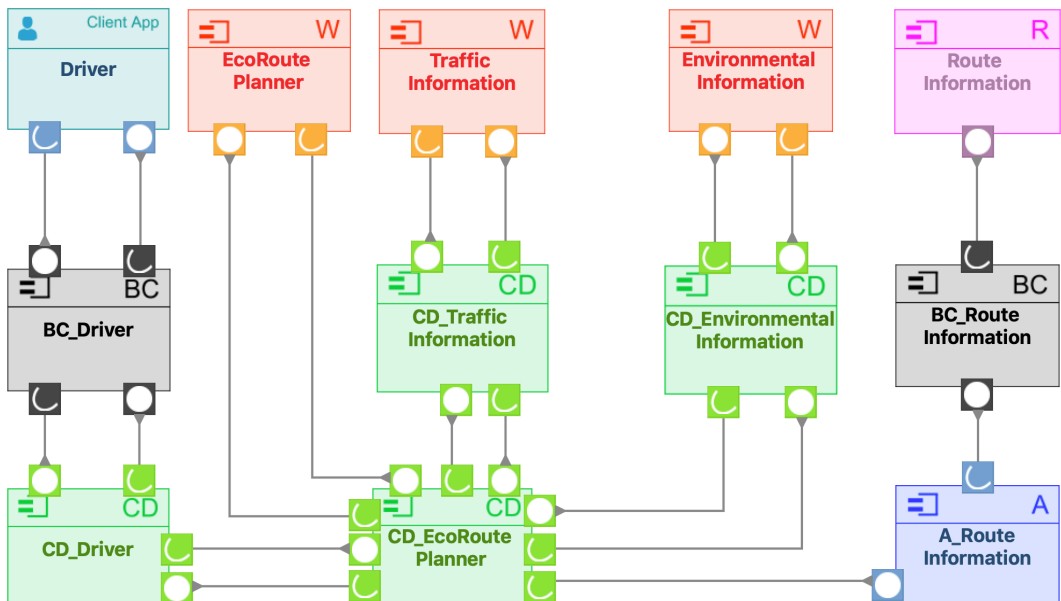

**Figure 7.** ECO Driving Planning architecture.

The system that realizes the Eco Driving Planning choreography (Figure 7) involves Coordination Delegates (CDs) for the following participants: `Driver`, `Eco Route Planner`, `Traffic Information` and `Environmental Information`. Each CD, except for the CD related to the driver client application, is connected with the web services implementing the business logic of the related participant. In particular, as explained in Section 4, the CDs and the web services for the participants `Traffic Information` and `Environmental Information` are connected in a bi-directional way to realize the event-based communication required when new information is available. Moreover, a binding component (BC with a black label) is needed for the driver and route information services. The latter also needs an adapter (with a dark blue label).

Figure 8 shows the software architecture of the system that realizes the Smart Data Collector choreography. It contains CDs for the following participants: `Viability Control Center`, `Smart Data Collector`, `Environmental Information`, `Weather`, `Air Quality`, `Roads Status`, `Traffic Information`, `Congestion` and `Accidents`. In particular, the CDs corresponding to the participants `Smart Data Collector` and `Environmental Information` are connected with the web services containing the related business logic. As said before, the accidents, congestion, weather, air quality and roads status services are MQTT services and hence they need a binding component. Moreover, weather, air quality and roads status also require an adapter.

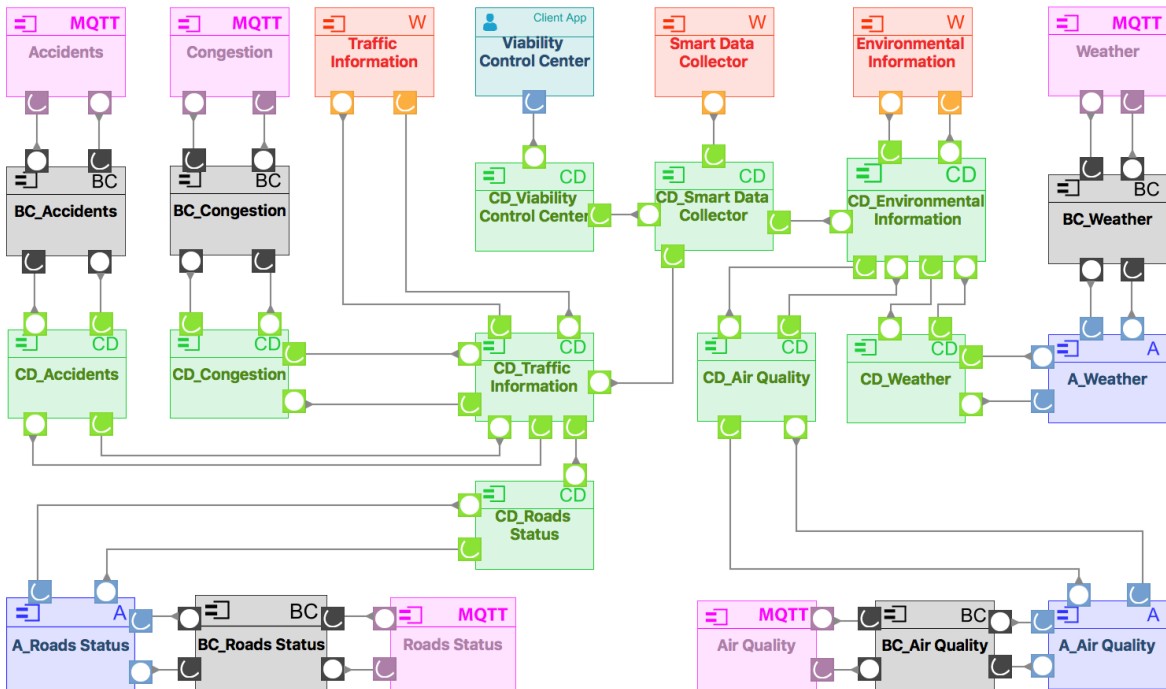

**Figure 8.** Smart Data Collector architecture.

### 4.2. Experimental Evaluation

Our approach has been evaluated by conducting an experiment for the CRMS use case introduced above. It is worth noticing that the work described in this paper is in the domain of software engineering for service-oriented distributed systems. Thus, the experiment that we carried out follows widely established guidelines for experimental software engineering that are described in [58]. The experiment that we describe in the following serves to evaluate the goodness of the CHOReVOLUTION approach with respect to using a traditional development method. This evaluation is based on comparing the development effort required by using the CHOReVOLUTION IDRE with the development effort required by manually implementing the entire CRMS choreography from scratch. The development effort is measured in terms of person hours (ph) of development activities. Thus, the main goal of the experiment is to assess the benefit of the proposed approach against a set of Key Performance Indicators (KPIs) and hence to measure the time saving for realizing the use case with our approach when compared to the development approaches our Master Degree students have been learning during software engineering courses. The considered KPIs are:

- **Effort for implementing the (possibly) distributed workflow (coordination logic)**: effort to develop the distributed coordination logic that guarantees the collaboration among the involved concrete services;
- **Effort for implementing the missing logic (business logic)**: effort to implement the missing business logic beyond the one implemented by the reused services;

- **Effort for integrating third-party or legacy services**: effort required to interconnect heterogeneous third-party or legacy services to fit the (possibly) distributed workflow specification.

The KPIs have been measured during the implementation phase (**Hypothesis**), which consists of the development of the choreography-based system from scratch. The time saving is measured in terms of person-hour (ph). In particular, regarding the CRMS use case, we employed the following two experimental units.

**Experimental unit 1:** *CHOReVOLUTION approach*—full usage of the CHOReVOLUTION IDRE except for the development of the mobile application, which is out of the scope.

**Experimental unit 2:** *General-purpose enterprise-oriented technology*—full usage of the technologies daily adopted by the students, i.e., JAX-WS and JAX-RS frameworks together with other supporting standards and libraries such as Java Persistence API, Jackson, etc., and Eclipse as IDE.

Regarding the efficient usage of the CHOReVOLUTION technologies within the experimental unit 1, the involved students spent six hours of training during dedicated lectures in a master degree course on advanced service-oriented software engineering.

The technologies of the experimental units 2 for the CRMS use case were selected considering that the students were already familiar and skilled with these technologies. It is clear that there exist many other equivalently powerful alternatives. However, opting for an alternative would have required a training effort that could not be afforded by the students because of exam time constraints. In any case, apart from time constraints and assuming the possibility to choose for an alternative, we can argue that, however (reasonably) long the training could last, it would be not easy to reach the same level of expertise, hence possibly compromising the validity of the experiment.

The considered KPIs are mapped to the following experimental tasks. In particular, the effort for implementing the distributed workflow is mapped to the **coordination logic** implementation, whereas the effort for implementing the missing logic is mapped to the **prosumer services** implementation. Finally, the effort for integrating third-party or legacy services is mapped to the **adaptation logic** implementation.

Both experiments were conducted by two different development teams. The team involved in the experimental units 1 was formed by two students, namely Dev1 and Dev2, engaged in the use cases development for the INCIPICT project. The other team, involved in the experimental units 2, was formed by the above two developers plus two others, namely Dev3 and Dev4. The experimental tasks were assigned to the students in order to eliminate the potential bias of person-task links.

Table 1 reports the experimental tasks assignment for the CRMS use case. In particular, Dev1 was assigned to the coordination and adaptation logic in the experimental unit 1. Dev2 was assigned to the prosumer services in the experimental unit 1. Dev3 was assigned to the coordination logic in the experimental unit 2. Finally, Dev4 was assigned to the prosumer services and the adaptation logic in the same experimental unit 2.

**Table 1.** Experimental tasks assignment CRMS use case.

| Experimental Tasks | Experimental Unit 1 | Experimental Unit 2 |
| --- | --- | --- |
| Coordination Logic | Dev1 | Dev3 |
| Prosumer Services | Dev2 | Dev4 |
| Adaptation Logic | Dev1 | Dev4 |

All students had equivalent professional skills, and familiarity with the BPMN2 choreography notation and the concerned technologies. All of them started performing their tasks out of already specified choreographies for the CRMS use case. Thus, the choreography specification phase was not part of the experiment.

**Hypothesis—** We found that the proposed approach significantly decreased the time required to implement the CRMS use case. Table 2 describes the activities performed within each experimental

unit for accomplishing the experimental tasks of the CRMS use case. It is worth noticing that the proposed approach provides a higher support to automation with respect to the other approach, which requires a manual implementation or a manual customization.

**Table 2.** CRMS experimental tasks activities.

| Experimental Tasks | Experimental Unit 1 | Experimental Unit 2 |
|---|---|---|
| Coordination logic | The code realizing the distributed coordination logic is automatically generated into a set of CDs, without requiring any manual intervention | Implement the coordination logic and the business logic for each of the distributed workflows. Design a mechanism to realize distributed workflow (subsequent tasks, parallel/decision gateways). Design a mechanism to handle asynchronous tasks in parallel. |
| Prosumer services | The skeleton code of the prosumer services is automatically generated. Thus, developers are required to only fill in the blanks of highlighted and partially ready pieces of code | The prosumer services are manually implemented. In particular, for each choreography task involving a specific participant, all the logic to manipulate received messages and to build the messages to be sent need to be coded from scratch. The developers have to maintain the data and message storage to ensure the messages are well parsed and routed through different distributed flows with no data lost. |
| Adaptation logic | Starting from an adapter model the adaptation logic is automatically generated into a set of ADs | The adaptation logic to bind the concrete services to the choreography participants in case of interfaces mismatches is manually implemented. |

For each experimental unit, Table 3 reports the person hours (ph) employed to carry out the experimental tasks of the CRMS use case, together with the total amounts of ph. In particular, the total amounts for the experimental unit 2 highlight in brackets the ph saved by using the proposed approach. The general-purpose enterprise-oriented approach took more than five times longer than the proposed approach.

**Table 3.** CRMS experimental tasks results—Implementation phase.

| Tasks | Experimental Unit 1 (ph) | Experimental Unit 2 (ph) |
|---|---|---|
| Coordination logic | 0 | 100 |
| Prosumer services | 12 | 22 |
| Adaptation logic | 12 | 28 |
| **Total** | **24** | **150 (126 saved)** |

This result together with the amount of ph saved in each experimental unit reveal that the proposed approach has great potential in developing choreography-based systems and the use case got a full benefit from it.

It is worth mentioning that the execution time of the CDs generation is polynomial in the number of tasks and gateways of the choreography specification [20]. Thus, for the experimental unit 1, the amount of ph for the generation of the coordination logic is considered to be equal to 0 since it takes just a few seconds of time to be performed. The amount of ph spent for the experimental unit 1, and hence the resulting amount of ph saved, do not consider the training time on CHOReVOLUTION technologies mentioned above. However, by even accounting for it, there would be a significant benefit anyway: 120 ph saved instead of 126 ph saved. The same applies for the automated generation performed by the tasks "Prosumer services" and "Adaptation logic".

## 5. Conclusions and Future Work

Since the earthquake of 6 April 2009, the city of L'Aquila has been involved in a massive and innovative reconstruction process. As a consequence, nowadays, the city can be exploited as a living laboratory model for applications within the context of smart cities.

In this context, the INCIPICT (INnovating City Planning through Information and Communication Technologies) project aims at creating a metropolitan network consisting of a fiber optic ring dedicated to the Public Administration, Schools, and Universities. This infrastructure enables the Next Generation Internet vision, and hence it can be exploited to provide innovative services for citizens.

By exploiting an automated synthesis approach for the production of service choreographies that we have implemented into the CHOReVOLUTION development platform, this paper discusses the realization of a choreography-based Collaborative Road Mobility System (CRMS) for the city of L'Aquila. The CHOReVOLUTION platform is an integrated platform for developing, deploying, executing and monitoring choreography-based distributed applications. The considered CRMS is one of the use cases belonging to the INCIPICT application scenarios. The CRMS allows vehicles and transport infrastructure to interconnect, share information and use it to coordinate their actions. Furthermore, it provides traffic coordination services exploited through a mobile app for assisting drivers in the most eco-friendly and comfortable driving experience.

By experimenting with the realization of the CRMS use case, we evaluated our CHOReVOLUTION synthesis approach against a set of Key Performance Indicators. The aim of the defined KPIs is to measure the time saving for realizing the CRMS use case by exploiting the automation facilities offered by the CHOReVOLUTION platform, compared to a more "traditional" development approach (i.e., without using the CHOReVOLUTION platform).

During the evaluation, the skilled Master Degree students involved in the experiment experienced a significant time decrease with respect to the development time required to develop the CRMS by exploiting the traditional development technologies and tools that they have learned during the software engineering courses offered by the Master Degree programme in Computer Science of the University of L'Aquila. In particular, the feedback they provided us with has shown that the advantages demonstrated in the use case, which have been tangibly measured in terms of time saving, comprises: reusing existing services, support for distributed composition, support for automation of coding operations and provision of correctness by construction. It is worth mentioning that these aspects constitute attractive factors that can effectively contribute to the success of the CHOReVOLUTION platform as an integrated development and runtime environment for effectively supporting the realization of the INCIPICT vision.

The outcomes of the experiment cannot be considered as applicable to any domain. Nevertheless, the extent to which the findings and the expected advantages can be generalized is relevant and interesting. Content integration, distributed workflow and business process management, especially applied to the Smart Cities context, are among the most evident scenarios to be exploited. Mobility-as-a-Service applications, which rely on the cooperation among stakeholders, public and private ITS organizations (public transport companies, parking companies, etc.) to offer static and dynamic information, are an example. More generally, ICT companies can benefit from the programming paradigm offered by the proposed approach in terms of software reuse, rapid prototyping and agile development.

As future work, we plan to conduct further experiments on a wider scale. To this direction, we will define more and suitable KPIs required to evaluate the performance of the choreography-based system (semi-)automatically generated by the CHOReVOLUTION IDRE with respect to the performance of the system manually implemented from scratch. This will consolidate the validation with stronger results regarding the effectiveness of our approach.

Furthermore, different research directions will be taken in the future to exploit the CHOReVOLUTION approach for the realization of INCIPICT use cases that deal with emergency management during and after earthquakes, disaster recovery, etc., based on technologies for structural building monitoring, building automation, smart evacuation, smart crowding routing, etc.

Finally, other future research actions concern understanding how the synthesized choreographies may exploit the services offered by the optical, or 5G, network layer (e.g., software defined networking, network slicing configuration) to generate more complex and reconfigurable application-level

coordination logic, hence enabling choreography dynamic evolution with respect to possible context changes.

**Author Contributions:** The authors contributed equally to this work.

**Funding:** This research was funded by the Ministry of Economy and Finance, Cipe resolution n. 135/2012.

**Acknowledgments:** This work is partially supported by: (i) the Ministry of Economy and Finance, Cipe resolution n. 135/2012 (INCIPICT), and (ii) the GAUSS national PRIN project (Contract No. 2015KWREMX).

**Conflicts of Interest:** The authors declare no conflict of interest.

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
