# Peer review of "A Choreography-Based and Collaborative Road Mobility System for L’Aquila City"

_futureinternet, doi:10.3390/fi11060132_

Round 1

Reviewer 1 Report

This paper introduced a meaningful and representative study case of choregraphy-based models on CRMS. Just a few suggestions.

1. the Section 6 "related work" should be moved to the beginning part of paper.

2. the Section  4 "The CHOReVOLUTION systhesis method at work on the CRMS use case" and Section "Experimental evaluation" should be merged into Section 3 as sub-parts.

3. In "Experimental evaluation", I suggest the authors give some illustration about the execution efficiency of the approach. If possible, compare it with the traditional models. 

Author Response

Our answers to the reviewer comments are reported in detail in the uploaded .PDF.

Reviewer 2 Report

It  is very important to define the technology used and why it was selected, together with the description of the mobility system in the region and specifically to see the costs of its implementation. The authors have previous research works so it is assumed that their implementation of the project has been in a scalable manner.

A real application is of utmost importance to achieve impact on society and potentiate the improvement of a component of a Smart City, the case study that they present, although it is described if they should focus on justifying the achievements that have been made in Consideration changes in scalability.

One aspect to consider is that a resource platform is created but it is not specified if any will be decisive for a similar situation after the earthquake.

I would have liked a description of their future work, because although they describe a technological niche, they do not adjust their focus on improving the population after suffering an earthquake, such as specialized services that should continue to work for patients and injured.

Explain how they came to evaluate with a more "traditional" model, as when they describe this at the end of their conclusions: "By experimenting with the realization of the CRMS use case, we evaluated our CHOReVOLUTION synthesis approach against a set of Key Performance Indicators".

This research is very interesting, but will be improved detail a few aspects of this research.

This research is very adequate, but will be improved using a correct design of experiments as in:

Augmented Reality Labels for Security Signs based on Color Segmentation with PSO for Assisting Colorblind People

MM Rivera, AP Díaz, JC Reich, JCP Gallegos, AO Zezzatti

International Journal of Combinatorial Optimization Problems and Informatics …

2019

Is very important describe more the porposed technolgy used as in:

Application of IoT with haptics interface in the smart manufacturing industry

RAC Masse, A Ochoa-Zezzatti, V García, J Mejía, S Gonzalez

International Journal of Combinatorial Optimization Problems and Informatics …

2019

And finally is very important describe better the concept of mobility system as in:

:
Smart Cities Concept: Smart Mobility Indicator. Cybernetics and Systems 50(2): 118-131 ()

Saud Althunibat, :

Random Waypoint Mobility Model in Space Modulation Systems. IEEE Communications Letters23(5): 884-887 ()

Author Response

(The authors gave the same response as above.)
